# Changes in the Structure and Mechanical Properties of the SAV-1 Alloy and Structural Fe-Cr-Ni Steels After Long-Term Service as Core Materials in Nuclear Reactors

**DOI:** 10.3390/ma18143391

**Published:** 2025-07-19

**Authors:** Alexey Dikov, Sergey Kislitsin, Boris Ivanov, Ruslan Kiryanov, Egor Maksimkin

**Affiliations:** Institute of Nuclear Physics, Ministry of Energy of the Republic of Kazakhstan, Almaty 050032, Kazakhstan; dikov@inp.kz (A.D.); kapjicohh@gmail.com (B.I.); e.maksimkin@inp.kz (E.M.)

**Keywords:** core materials, mechanical tests, creep tests, WWR-K reactor, fast neutron BN-350 reactor, neutron irradiation

## Abstract

This article presents the results of studies of the degradation of the structure and mechanical properties of the core materials BN-350 fast neutron and research WWR-K reactors required to justify the service life extension of early-generation power and research reactors. Extending the service life of nuclear reactors is a modern problem, since most operating reactors are early-generation reactors that have exhausted their design lifespan. The possibility of extending the service life is largely determined by the condition of the structural materials of the nuclear facility, i.e., their residual resources must ensure safe operation of the reactor. For the SAV-1 alloy, the structural material of the WWR-K reactor, studies were conducted on witness samples which were in the active zone during its operation for 56 years. It was found that yield strength and tensile strength of the irradiated SAV-1 alloy decreased by 24–48%, and relative elongation decreased by ~2% compared to the unirradiated alloy. Inside the grains and along their boundaries, there were particles of secondary phases enriched with silicon, which is typical for aged aluminum alloys. For irradiated structural steels of power reactors, studied at 350–450 C, hardening and a damping nature of creep were revealed, caused by dispersion hardening and the Hall–Petch effect.

## 1. Introduction

During nuclear reactor operation, the materials and components of the core are exposed to various factors that lead to deterioration and degradation of their properties. This limits the lifetime of the reactor. In most cases, replacing the components of the reactor core is not rational or even possible. However, there are some examples of the application of special procedures to restore the properties of the core materials, in particular recovery annealing of the reactor vessel. Such work has been carried out to extend the service life of WWER-440 reactors in various countries [1,2]. For research reactors (RRs), the problem of aging of the core materials is especially acute, since according to the IAEA Research Reactor Database, more than half of the reactors currently in operation are over 50 years old [3].

There are several methods for assessing the possibility of extending the service life of various structures subject to constant or changing stresses, external and internal influences, environmental factors, etc. [4,5]. Extending the service life of RRs is possible after a risk analysis, based on data on the properties of the core materials exposed to neutron irradiation, temperature and water-chemical conditions, and other factors. An authorized expert commission is assessing the possibility of extending the operation of power and research nuclear facilities following the recommendations of the IAEA [6].

In-reactor and out-of-reactor studies are used to assess the impact of various factors that influence the properties of materials, for example, irradiation with high-intensity neutron fluxes to simulate radiation damage during the planned service life of the material. In [7], a new type of additive manufacturing material based on austenitic stainless steel 316 L was studied, including post-irradiation material mechanical properties analysis. It is known that neutron irradiation is an expensive and time-consuming process, especially considering higher fluxes and fast neutron spectra. Some arguments can be made to simulate neutron irradiation with ion irradiation [8]. It is relatively faster to obtain a desirable dpa and a similar resulting microstructure by ion irradiation [8]. It was shown that it is possible to establish a relationship between the microstructure and mechanical properties of structural materials [9]. Still, some broader understanding of both methods of irradiation should be made to implement this method.

The most representative data on properties of nuclear materials are obtained by the analyses of the witness samples placed in the reactor core at its commissioning and samples of removed/replaced elements of the core components.

In Kazakhstan, the WWR-K reactor (Almaty) has been in operation since 1967 [10,11]. From 2015 to 2016, the reactor underwent conversion from high-enriched to low-enriched fuel [10,12]. The main material of the core is the SAV-1 alloy; the closest analogues are aluminum alloys of the 6XXX series. The support grid, shell, inner part of the vessel, and other elements of the core are made of SAV-1. The SAV-1 alloy was hardened and aged by heating to 520 °C in air. After such processing, the alloy has a structure typical of a low-alloy aluminum composition with dispersion strengthening and uniform distribution of the magnesium silicide phase [13].

It is well established that aluminum alloys are subject to radiation hardening and embrittlement [13,14,15]. The main contribution to these processes is caused by displacement damage and transmutation that lead to silicon isotopes accumulation by Reactions (1) and (2).(1)Al (n,γ)  27Al 28(2)Al 28 → Si 28+β

The accumulation of silicon isotopes, depending on the fluence and neutron spectrum, can make a greater contribution to the change in the properties of the material than the gaseous products of transmutation by the reactions (n, α) and (n, p), which is a distinctive feature of aluminum alloys [13].

In 1999, the BN-350 reactor (USSR designed and produced located in Kazakhstan, Aktau) was shut down and is currently undergoing a phased decommissioning process [16]. In the core of the reactor, austenitic steels were widely used.

Austenitic steels are characterized by radiation hardening [17]. Studies of steels irradiated in the BN-350 reactor are primarily aimed at substantiating the possibility of extending the service life of water-cooled power reactors. Knowledge of the degradation of the properties of materials irradiated in the neutron spectrum of a fast reactor allows one to use a conservative approach when assessing the deterioration of structural materials used in water-cooled reactors with a thermal neutron spectrum.

## 2. Materials and Methods

### 2.1. Sample Irradiation Conditions

SAV-1. In 1967, before the start-up of the reactor, witness samples of the SAV-1 alloy were placed in the active zone to assess the deterioration of the physical and mechanical properties of the reactor material during operation. The location of the samples in the reactor core is marked with a red dot in Figure 1. The witness samples were removed from the reactor in 2024. The total irradiation time was ~56 years, in water, at an average temperature of 80 ± 5 °C.

During the irradiation, the radiation damage dose to the SAV-1 witness samples was ~3.8 dpa. The radiation damage dose was calculated by the Monte Carlo method by the MCNP6 transport code; EFF-4T1, ENDF/B-VII.1, ENDF/B-VIII, JENDL-5, TENDL-2021 nuclear constant libraries [18,19,20]; and the NRT model [21]. The displacement cross-sections for SAV-1 used in the calculation were obtained by the Karlsruhe Institute of Technology [22]. The composition of the SAV-1 alloy is presented in Table 1. The samples for the mechanical test of the SAV-1 alloy had the shape of a double blade with the working part dimensions of 22 × 4 × 0.5 mm.

Austenitic steels. The second group of samples was made of 0.12C18Cr10NiTi steel and 0.08C16Cr11Ni3Mo steel (made in USSR) (analogous to AISI 316) cut from the edges of the spent fuel assemblies of the BN-350 reactor. The composition of the steels is presented in Table 2.

The sample cutting scheme is shown in Figure 2. Calculations of temperature–dose characteristics of steel samples are given in Table 3.

The steel samples were plates measuring 20 × 2 × 0.3 mm, with a working part length of 10 mm. The samples were cut on a PTF-2 electrical discharge machine (made in USSR) in distilled water in a cross-section relative to the fuel assembly axis. Then, the surface of the samples was processed by mechanical grinding on sandpaper of different grain sizes. This procedure allows for removing the surface layer of steel, which is modified during electrical discharge cutting.

### 2.2. Mechanical Tests

Tensile tests were carried out on an LR5K Plus universal testing machine (Lloyd Instruments, Bognor Regis, UK) with a crosshead speed of 0.5 mm/min. The test temperature was 90 °C for the SAB-1 and 20, 350, and 450 °C for steel. The applied loads and deformations were recorded continuously throughout the experiment. The limits of the permissible relative error of the force and deformation sensors were 0.5%, and the discreteness of the digital reading device was 0.005% of the nominal load of the force sensor. The results were analyzed using NEXYGEN Plus software 3.0. The test temperature was recorded and controlled using a VRT-1 high-precision temperature controller (Autonics Corporation, Seoul, Republic of Korea) equipped with a type K (chromel-alumel) thermocouple; the error in determining the temperature of the working part of the sample did not exceed ±2 °C.

The choice of creep test conditions (temperature, load) was determined based on the operating conditions of the reactor core elements. Under standard operating conditions of the WWR-K reactor, the temperature heating of the core elements does not exceed 75 °C (short-term heating up to 90 °C is possible), and the stresses arising in the materials of the loaded core elements do not exceed 50 MPa. Therefore, creep tests of the SAV-1 alloy were conducted at a temperature of 90 °C and a load of 100 N (which corresponds to 45 MPa).

The conditions for testing the creep of steels (BN-350 reactor materials) were as follows: temperatures of 350 °C (average operating temperature) and 450 °C (critical operating temperature) and loads of 100 N (corresponds to a working stress of 150 MPa) and 450 and 470 N (stresses close to the conditional yield strength of steels).

Before testing, the samples were heated without load. The heating rate was 7 °C/min. Samples were kept at a given temperature for 30 min before the loading. The rate of extension to the specified load values was 0.5 mm/min. The duration of creep tests was from 600 to 1600 h, and the samples were not brought to failure. Due to the long duration, in some cases the termination of experiments was due to external factors (schedule of shift work of the reactor, scheduled preventive maintenance, etc.).

The microstructure and fracture surface of steel samples were studied using an Axio Observer optical microscope (Carl Zeiss Microscopy GmbH, Jena, Germany) and a Hitachi TM 4000 Plus scanning electron microscope (Hitachi High-Tech Corporation, Tokyo, Japan). The microstructure was revealed by chemical etching in a solution of HNO_3_ + H_2_SO_4_ + FeCl_3_.

## 3. Results and Discussion

### 3.1. SAV-1

Figure 3 shows the tensile diagram of the SAV-1 alloy (non-irradiated and irradiated to 3 dpa). The mechanical characteristics are given in Table 4. It can be seen from Figure 3 that the irradiated alloy is sufficiently plastic, and no critical decrease in relative elongation was detected in the irradiated SAV-1 alloy. The relative elongation of the irradiated SAV-1 alloy, in comparison with the non-irradiated alloy, decreased nonsignificantly by ~2%. The duration of the plastic deformation section in the tensile diagram indicates that long-term irradiation of the witness samples did not cause embrittlement of the material. At the same time, the decrease in yield strength (Ϭ_0.2_) and tensile strength (Ϭ_B_) by 24 and 48%, respectively, indicates weakening of the alloy.

The fracture character for both non-irradiated and irradiated alloys is viscous (Figure 4). The fracture surface has a pitted structure. A distinctive feature of the fracture surface of the irradiated alloy is the presence of particles (~3 µm) in the pits (Figure 4c). No particles were recorded in the fracture of the non-irradiated alloy (Figure 4d). According to the energy-dispersive analysis, the particles were enriched with silicon, oxygen, and magnesium. Similar particles in the structure of SAV-1 were recorded by other authors, for example, in the work [14]. In [15], it was found that irradiation of the SAV-1 alloy leads to the formation of Mg_2_Si phase precipitates. In these works [14,15], the authors investigated the control rod of the core of the WWR-K reactor. An analysis of the samples of the alloy SAV1 subjected to neutron irradiation to a fluence of 10^26^ n/m^2^ was carried out. It was shown that with an increase in fluence, the alloy exhibited radiation hardening. Subsequent annealing of irradiated SAV-1 at a temperature of 100–300 °C led to a decrease in the hardness of the material.

It should be noted that the material of a control rod from the core was exposed to higher neutron fluences during operation. Witness samples studied in this work were irradiated with a very slow rate of damage dose accumulation of 3 dpa over 56 years at a temperature of 80 °C. The presence of large particles of the strengthening phase in the alloy fracture (typical of aged aluminum alloys) and a decrease in mechanical properties indicate aging of the material. Thus, neutron irradiation with a very slow rate of damage does not lead to the usual effects of radiation exposure. The impact of this irradiation mode resembles long-term thermal aging of the material.

Figure 5 shows the creep rate of the SAV-1 alloy after long-term reactor irradiation. It is evident that after reaching the specified test conditions, the alloy had a relatively high creep rate, which faded out over time. Such behavior of the creep curve is typical for tests in the region of relatively low temperatures and stresses. Upon reaching the specified test conditions, a constant increase in stress occurred in the material, which stimulated the nucleation and movement of dislocations. After that, the growth of stress stopped, and the inertial movement of dislocations continued until the local stresses dissipated. At the same time, some of the dislocations remained in a stressed state, stuck on obstacles, and relaxed. This can be seen in the diagram as a transition stage. Further movement of dislocations, at the steady-state stage of creep, occurred mainly by the “creep” mechanism, i.e., a change in the slip plane. The change in creep rate at different time intervals was due to the presence of dislocation stoppers of different types in the structure of the material.

Considering the low rate of the damage dose, the significant plasticity of the alloy after prolonged reactor irradiation, and the damping nature of creep under temperature and force conditions typical for operating conditions, it can be concluded that the alloy is capable of further use.

### 3.2. BN-350 Structural Steels

The mechanical characteristics of the examined steels were determined, and their dependencies on the test temperatures were made according to the results of the uniaxial tension tests (Figure 6) [23]. The provided dependencies show that an increase in the test temperature from 20 to 350 °C reduced the strength characteristics, which does not contradict the known concepts [17,24,25]. However, at the same time, the relative elongation decreased, although, as is well known, plastic properties of steels increase at temperatures of 300 °C and higher [26]. However, the opposite effect for both steels with increasing of test temperature up to 450 °C was observed in the increase of yield strength and tensile strength and the continued decrease in elongation. These results can be compared with properties of unirradiated steels presented in [27].

In order to identify the causes of the strengthening effect, the microstructure of the steel samples was studied before and after mechanical testing. The fracture surface of the tested samples was also studied. The microstructure of the samples after testing are presented in Figure 7 and Figure 8.

The microstructure of 0.08C16Cr11Ni3Mo and 0.12C18Cr10NiTi steels after operation and subsequent “wet” storage in the storage pool was characterized by the various-grained structure. On the microstructure images, presented in Figure 7a,b and Figure 8a,b, it can be seen that the various-grained structure was retained after testing of both steels. Testing of 0.12C18Cr10NiTi steel at 350 and 450 °C (Figure 7a,b) resulted in coagulation of carbides concentrated at grain boundaries, including the regions of triple junctions. Additionally, at a temperature of 450 °C, a large number of microscopic particles formed in the steel. This is indicated by the appearance of ridges of separation from microscopic pits (Figure 7d). The nature of the fracture (i.e., viscous failure) stayed relatively the same up to a temperature of 350 °C. Next, the mechanism of fracture at 350 °C started to vary. This is evidenced by the formation of shallow pits and partitions between them in the deformation process (Figure 7c). At a temperature of 450 °C, the local porosity of steels increased, which is manifested in the appearance of a network of separating ridges of microscopic pits and the presence of secondary pits on the surface of large voids. The heterogeneity of the material structure led to the formation of a local stress gradient [28,29]. These additional stresses associated with local heterogeneity of the material structure led to a decrease in mechanical (strength) properties and earlier failure (reduction in relative elongation). Apparently, the strengthening of steel at a temperature of 450 °C was due to the inhibition of dislocations on a variety of newly formed particles.

A similar strengthening effect was also observed in irradiated 0.08C16Cr11Ni3Mo steel (Figure 8a–d). The testing at a temperature to 450 °C resulted in a great decrease in the number of large (>30 µm) grains in the fracture area. This was accompanied by an increase in the number of fine carbides (in the form of chains) in the grain bodies and on their boundaries. Also, traces of twinning are clearly visible in the images.

Figure 9 and Figure 10 show the dependences of the change in creep rate on time for austenitic steels under temperature and force conditions typical for operation in a reactor. Three main conditions are considered: 1—standard operating conditions of steels in a reactor (P = 100 N, T = 350 °C), 2—increase in operating temperature (P = 100 N, T = 450 °C), and 3—increase in mechanical stress levels (P = 450 ÷ 470 (~Ϭ_0.2_), T = 350 °C).

From Figure 9, it is evident that the creep of 0.12C18Cr10NiTi steel under the conditions of standard operation (100 N, 350 °C) is of a staged, damping nature, as shown by the red curve in Figure 9. The damping is manifested as a decrease in the creep rate at the steady-state stage.

At the increased temperature (450 °C) and the constant load (100 N), creep also had a damping character, as shown by the green curve in Figure 9. The creep rate at the initial stage was raised by about three times. Increasing the load to values close to the conditional yield strength of steel (P~Ϭ_0.2_), at a constant test temperature of 350 °C, increased the duration of the initial stage, as shown by the blue curve in Figure 9. The maximum of the creep rate at the initial stage decreased. At the steady-state stage, the creep proceeded at a relatively constant rate compared to standard operating conditions.

The creep rate values of 0.08C16Cr11Ni3Mo steel under normal operating conditions were similar to those of 0.12C18Cr10NiTi steel. See the red curves in Figure 9 and Figure 10.

The creep rate at the initial stage also increased at an increased temperature of 450 °C and a constant load of 100 N, as shown by the green curve in Figure 10. However, this increase was less significant (1.5 times) than for 0.12C18Cr10NiTi steel. At the steady-state stage, the creep rate of 0.08C16Cr11Ni3Mo steel decreased to values close to creep under normal operating conditions, which was not observed for 0.12C18Cr10NiTi steel.

The creep of 0.08C16Cr11Ni3Mo steel, under loads close to the conditional yield strength (P~Ϭ_0.2_) and a test temperature of 350 °C, differed significantly from that of 0.12C18Cr10NiTi steel under similar conditions, as shown by the blue curves in Figure 9 and Figure 10. The initial stage of creep of 0.08C16Cr11Ni3Mo steel was not long and was characterized by a higher creep rate. With the transition to the steady-state stage, the creep rate faded to values close to creep under normal operating conditions and then proceeded at a relatively constant rate.

The duration and rate of creep at different stages were strongly influenced by the microstructure of the material and its development under changing external conditions. The studied steels had an austenitic microstructure with dispersion hardening (Figure 11). Dispersion hardening was provided by carbides of the MC and M23C6 types for 0.12C18Cr10NiTi steel and by carbides of the M23C6 type for 0.08C16Cr11Ni3Mo steel.

Figure 11a–c show the microstructure images of the tensile region of 0.12C18Cr10NiTi steel samples after creep testing. It is evident that for 0.12C18Cr10NiTi steel after creep testing under standard operating conditions (P = 100 N, Tisp = 350 °C), the presence of large (~4 μm) carbide particles M23C6 located along the boundaries and at the junctions of three or more austenite grains is characteristic (see Figure 11a). More dispersed carbides of the MC type (~1 μm) were dispersed in the body of the austenite grains. The austenite grains have sizes of 15–34 μm.

At the increased temperature (P = 100 N, Tsp = 450 °C), thickening of triple joints and cleaning of grain boundaries from M23C6 carbides were observed (Figure 11b). The concentration of MC-type carbides in the grain body was maintained. The size of austenite grains did not change significantly and was 15–47 μm. Apparently, the reduction in the number of large carbides (along grain boundaries), which act as dislocation stoppers, contributed to an increase in the creep rate, which is observed on the green curve in Figure 9.

At the elevated levels of mechanical stress (P~Ϭ_0.2_, T = 350 °C), austenite grains averaged to a size of 20 ± 2 μm. Weakly distinguishable, isolated, finely dispersed carbides measuring 4 ± 1 μm were observed along the grain boundaries. Large 8 ± 3 μm carbide formations were present in the body of some grains. However, most grains, primarily small ones, were free of precipitates.

In steel 0.08C16Cr11Ni3Mo, in addition to dispersion strengthening, deformation strengthening processes occurred. This is confirmed by the presence of twins of austenite grains in Figure 8 and Figure 11d–f. A large number of dispersed carbides in the steel structure and the presence of twins of austenite grains together provided more effective braking of dislocations. This is the reason for the higher heat resistance of steel 0.08C16Cr11Ni3Mo in comparison with steel 0.12C18Cr10NiTi.

The structure of 0.08C16Cr11Ni3Mo steel was heterogeneous, with grain sizes of 10–83 μm at loads of 100 N and a temperature of 350 °C. Dispersed carbides were mainly arranged in chains in the grain and along its boundaries. The distribution of carbides was uneven; grains free of carbides are observed in Figure 11d. These were mainly large grains of 50 μm and more.

Increasing the test temperature to 450 °C stimulated the growth of carbides in small austenite grains. Large austenite grains remained free of carbides. Small grains, up to 50 μm, were grouped. The grain size heterogeneity of the steel was maintained at 10–83 μm. The unevenness of the structural components contributed to the increase in the creep rate of 0.08C16Cr11Ni3Mo steel, and the large number of carbides ensured its insignificance at the initial stage due to the containment of dislocation.

The elevated levels of mechanical stress (P~Ϭ_0.2_, T = 350 °C) resulted in a decrease in the grain size to 50 ± 5 μm. A large number of grain clusters with a relatively small size of 12 ± 5 μm were observed. Dispersed carbides were arranged in chains in austenite grains. In some areas, carbide chains were not interrupted by grain boundaries but passed to the adjacent grain. Apparently, the damping of the creep rate at the steady-state stage, under these test conditions, is caused by grain refinement [30].

## 4. Conclusions

From the conducted study of changes in the structure and mechanical properties of the SAV-1 alloy and structural Fe-Cr-Ni steels as a result of long-term operation as a core material of nuclear reactors, the following conclusions can be drawn.

### 4.1. SAV-1 Alloy—A Structural Material for Research Nuclear Reactors

Long-term irradiation of the SAV-1 alloy with a low rate of damage dose accumulation (~56 years, dose ~3 dpa, 80 °C) leads to aging of the material, without manifestation of radiation effects. Yield strength values of the irradiated SAV-1 alloy are close to those for non-irradiated alloys. Relative elongation decreases by ~2%. Inside the grains and along the boundaries, there are particles of secondary phases enriched with silicon, which is typical of aged aluminum alloys.

It was shown that after active operation for 56 years, the resource of the aluminum alloy in terms of mechanical characteristics, such as strength, brittleness, and plasticity, is far from exhaustion. This is largely due to the low level of radiation damage to the non-replaceable devices, like the reactor vessel. The neutron radiation dose for the entire period did not exceed 5 dpa. An important factor limiting the resource, in addition to the degradation of the mechanical properties of structural materials, is corrosion damage [31,32]. For the WWR-K reactor, significant corrosion damage, such as pitting corrosion, was not noted. The above research results serve as one of the most important reasons for extending the active operation period of the reactor.

### 4.2. Fe-Cr-Ni Steels—A Material for the Ducts of Fuel Assemblies of the BN-350 Fast Reactor

In irradiated austenitic steels 0.12C18Cr10NiTi and 0.08C16Cr11Ni3Mo, with an increase in temperature from 350 to 450 °C, an increase in the yield strength and tensile strength was observed with a simultaneous decrease in relative elongation. These changes in properties are caused by processes of dispersion hardening, strain hardening, and grain refinement (Hall–Petch effect) occurring at a given temperature.

Structural changes in austenitic steels occurring during the creep process are similar to changes during uniaxial tension tests under identical temperatures.

The creep of irradiated reactor steels at temperatures and stresses typical for the operation of nuclear reactor core elements are characterized by stages: initial, transition, and steady-state creep were distinguished. At the steady-state stage, creep is of a damping nature. Damping is manifested by a decrease in the creep rate. The duration of stages and the creep rate at each stage are affected by both the initial microstructure and the tendency of the microstructure to restructure when external conditions change. Thus, it has been shown that an increase in the level of applied stresses to values close to the conditional yield strength leads to grain refinement, which increases the creep resistance of 0.08X16X11H3M steel.

Radiation effects in the steels are significantly more complex compared to the alloy SAV-1 due to a more complex composition and more severe irradiation conditions in a power reactor. In these materials, along with the formation of radiation defects and their agglomeration, segregation processes and phase transformations actively occur. The above results show that, despite the partial degradation of physical and mechanical properties, these materials withstand high levels of damage doses and retain their performance characteristics, which allows for extending the active service life of nuclear power plants. A distinctive feature of the results is that they were obtained for the steels used in the core of the BN-350 reactor. A characteristic feature of the BN-350 reactor is the low temperature of the coolant (liquid sodium): the temperature at the inlet is 280 °C, at the outlet is −400 °C, and the temperature in the center of the core is 350 °C. This means that the operating temperature in the core of the BN-350 reactor coincides with the operating temperatures of the coolant of the PWR, BWR, and WWR light water reactors. This allows the obtained data to be used to support and justify extending the service life of light water reactors.

## Figures and Tables

**Figure 1 materials-18-03391-f001:**
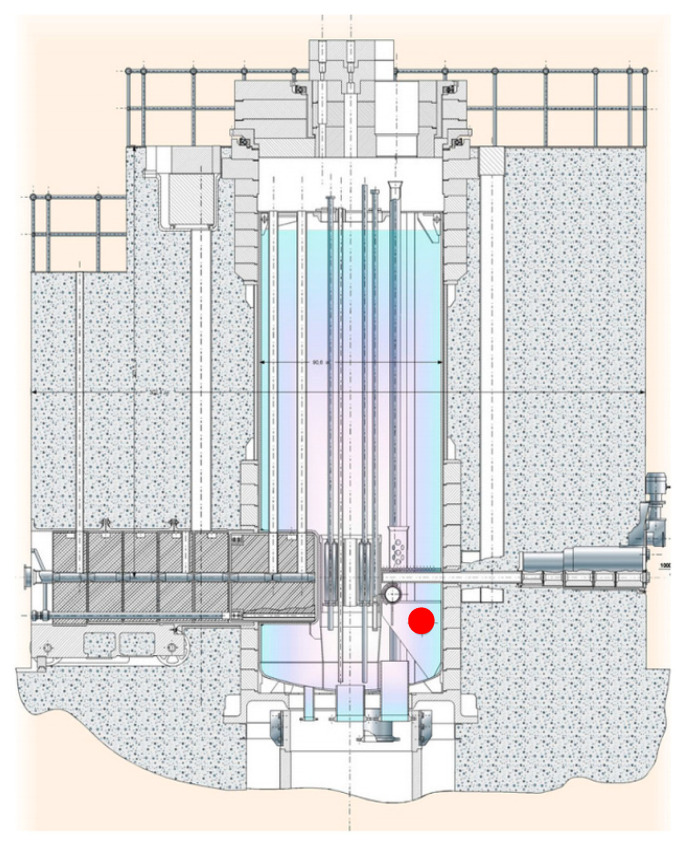
Vertical section of the WWR-K reactor. The inner diameter of the reactor is 4.8 m. Red dot showing a location of the samples during a reactor functioning.

**Figure 2 materials-18-03391-f002:**
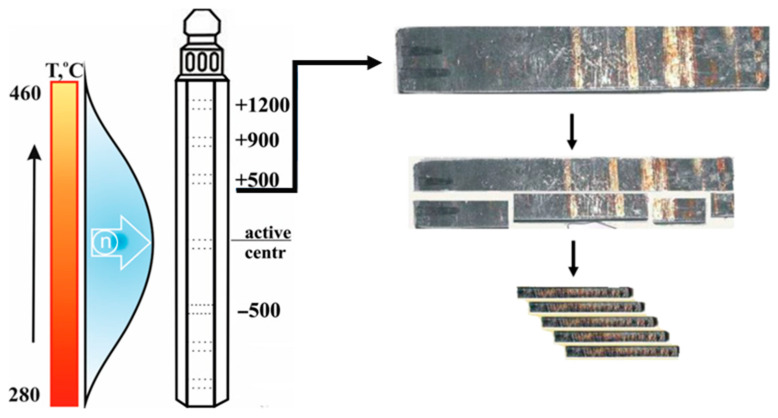
Temperature and neutron flux gradient and samples cutting diagram.

**Figure 3 materials-18-03391-f003:**
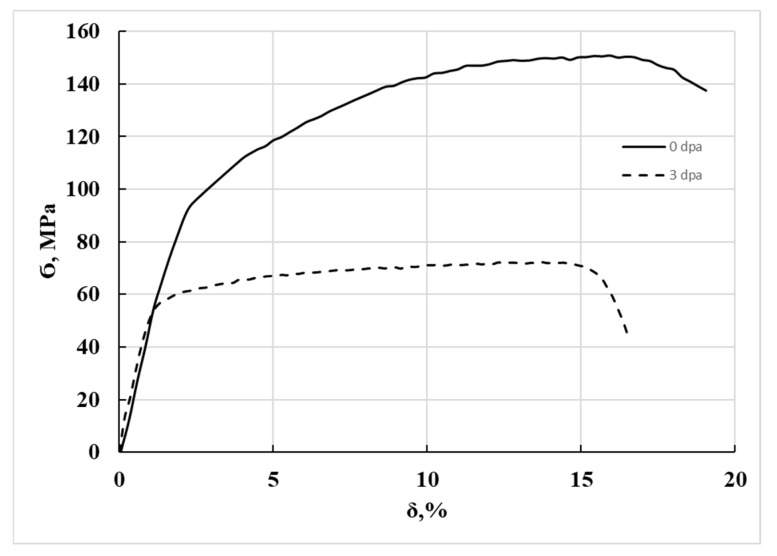
Stress-strain diagram of the SAV-1 alloy, non-irradiated (0 dpa) and after long-term exploitation (~3 dpa).

**Figure 4 materials-18-03391-f004:**
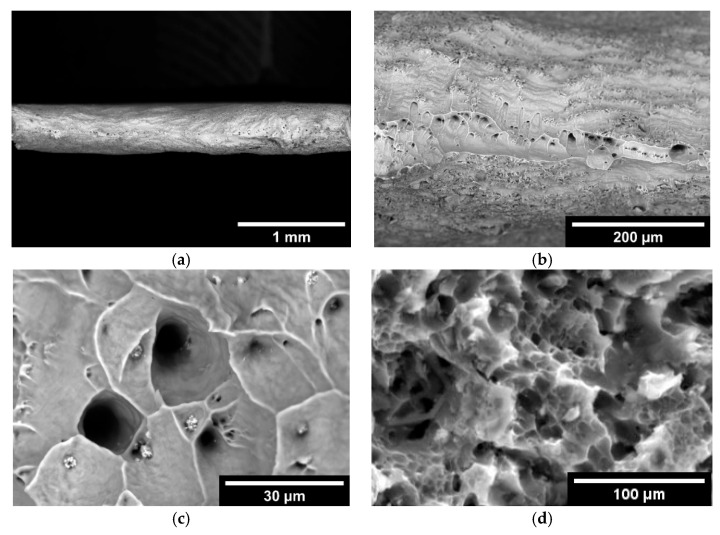
Fracture surface of irradiated (**a**–**c**) and non-irradiated (**d**) SAV-1 alloy after mechanical testing.

**Figure 5 materials-18-03391-f005:**
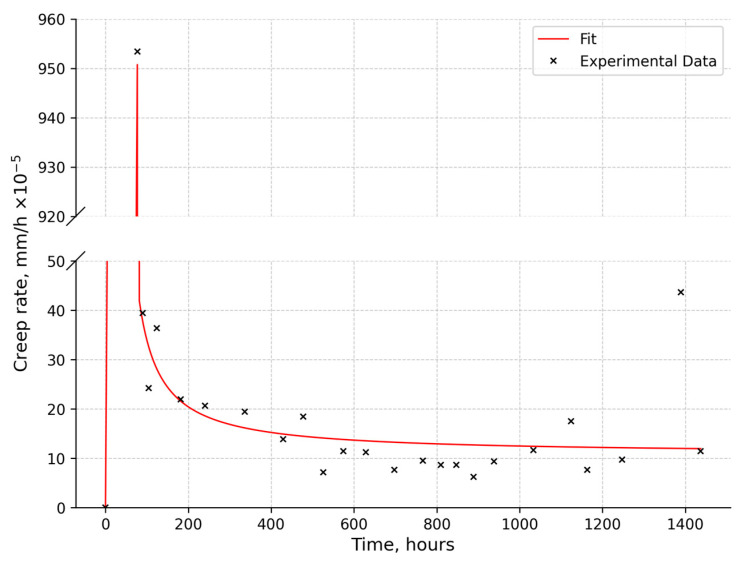
SAV1 irradiated to a 3 dpa creep rate (90 °C).

**Figure 6 materials-18-03391-f006:**
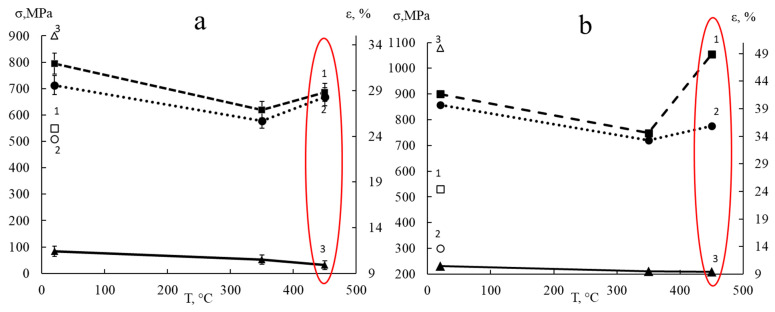
Changes in tensile strength (1), yield strength (2), and relative elongation (3) of irradiated 0.12C18Cr10NiTi steel (**a**) and 0.08C16Cr11Ni3Mo steel (**b**) depending on test temperature. Open symbols correspond to the values of ultimate strength (1), yield strength (2), and relative elongation (3) for non-irradiated steels obtained in mechanical tests at room temperature [23].

**Figure 7 materials-18-03391-f007:**
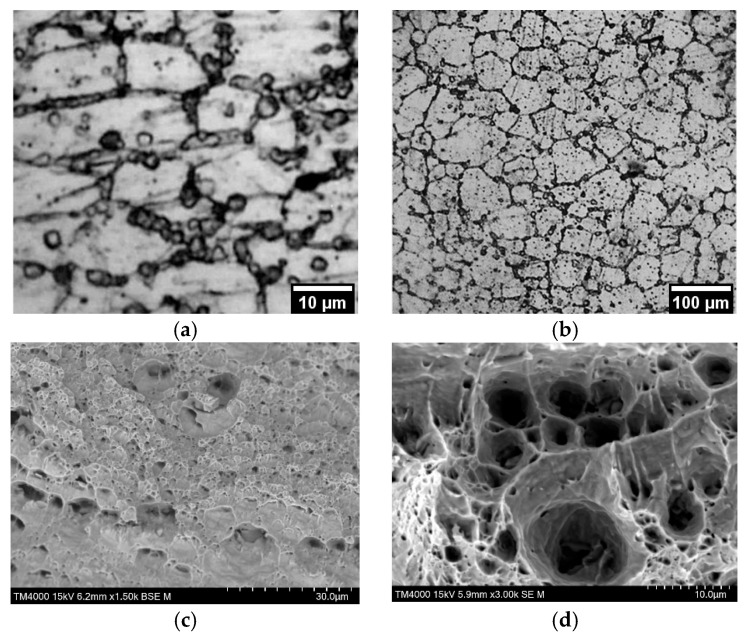
Microstructure (**a**,**b**) and fracture surface (**c**,**d**) of irradiated 0.12C18Cr10NiTi steel after mechanical tests at 350 (**a**,**c**) and 450 °C (**b**,**d**).

**Figure 8 materials-18-03391-f008:**
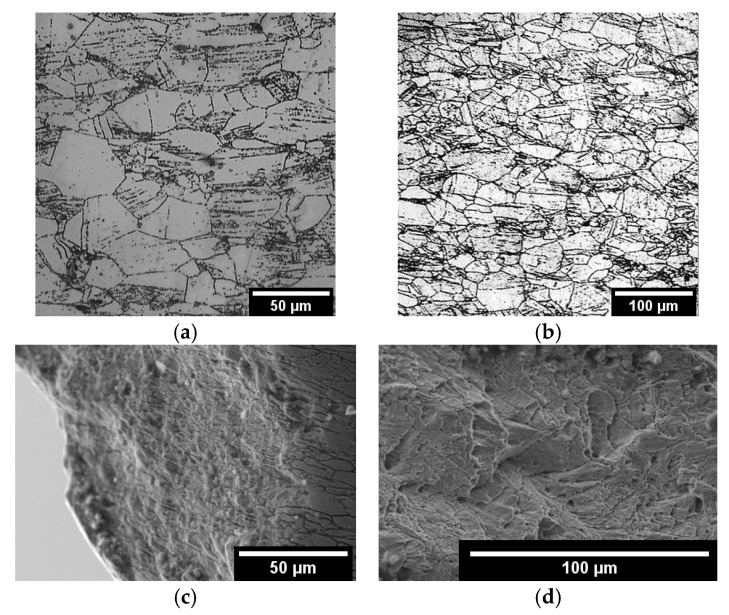
Microstructure (**a**,**b**) and fracture surface (**c**,**d**) of irradiated 0.08C16Cr11Ni3Mo steel after mechanical tests at 350 (**a**,**c**) and 450 °C (**b**,**d**).

**Figure 9 materials-18-03391-f009:**
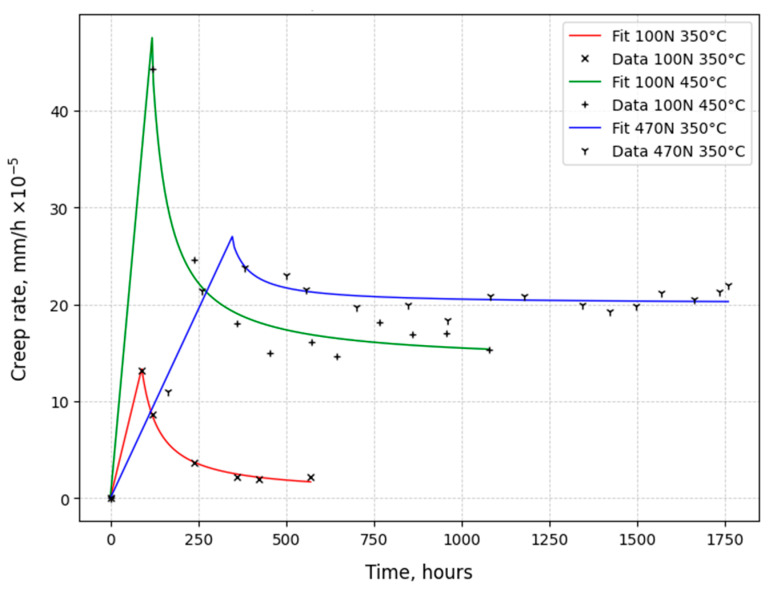
Creep rate of 0.12C18Cr10NiTi steel irradiated to 3 dpa.

**Figure 10 materials-18-03391-f010:**
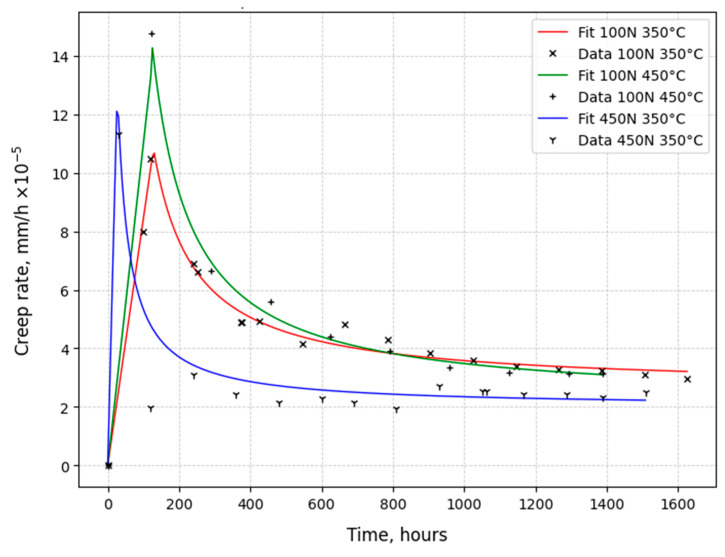
Creep rate of 0.08C16Cr11Ni3Mo steel irradiated to 6 dpa.

**Figure 11 materials-18-03391-f011:**
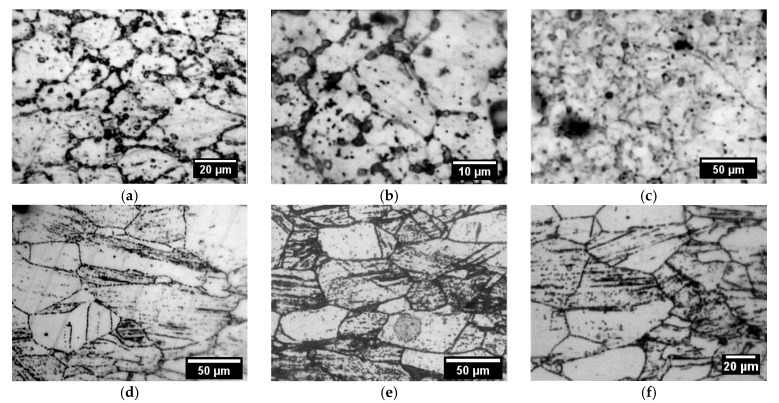
Microstructure of steels 0.12C18Cr10NiTi (**a**–**c**) and 0.08C16Cr11Ni3Mo (**d**–**f**) after creep tests. (**a**,**d**)—P = 100 N, T = 350 °C; (**b**,**e**)—P = 100 N, T = 450 °C; (**c**,**f**)—P~Ϭ_0.2_, T = 350 °C.

**Table 1 materials-18-03391-t001:** Composition of SAV1 alloy.

Element	Al	Si	Mg	Rest
Content	Base	0.7–1.2	0.045–0.9	≤0.3

**Table 2 materials-18-03391-t002:** Austenitic steels content.

Element	Fe	C	Cr	Ni	Ti	Mo
0.12C18Cr10NiTi	Base	0.12	17.0	10.66	0.5	-
0.08C16Cr11Ni3Mo		0.08	16.0	11.1	-	2.7

**Table 3 materials-18-03391-t003:** Austenitic steels irradiation dose and environment.

Material	Irradiation Dose, dpa	Mark, mm from AZ Center	Dose Rate × 10^−8^, s^−1^	Irradiation Temperature, °C
0.12C18Cr10NiTi	2–3	+900	0.5	400
0.08C16Cr11Ni3Mo	6	+500	1.9	365

**Table 4 materials-18-03391-t004:** Mechanical characteristics of the SAV-1 alloy before and after irradiation.

Irradiation Dose, dpa	Irradiation Temperature, °C	Ϭ_B_, MPa	Ϭ_0.2_, MPa	δ, %
0	90	150 ± 5	90 ± 5	18
~3	90	77 ± 5	68 ± 5	15 ± 2

## Data Availability

The original contributions presented in this study are included in the article. Further inquiries can be directed to the corresponding author.

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
