# Peer review of "Changes in the Structure and Mechanical Properties of the SAV-1 Alloy and Structural Fe-Cr-Ni Steels After Long-Term Service as Core Materials in Nuclear Reactors"

_materials, 2025, doi:10.3390/ma18143391_

Round 1
Reviewer 1 Report
Comments and Suggestions for Authors
Very good paper. The paper doesn’t contain any critical errors. The applied methodology is robust. The rationale for the research is obvious. The obtained results are credible. Neither improvements nor corrections are necessary. The main advantage of the paper is its concrete and compact character. Very good piece of scientific work.
Minor corrections:
Figure 1 – please add reference dimension.
Figure 3 – MPa
More references about methods for calculating neutron fluence, reaction rates, and finally dpa in burnup function can be added to provide more wider context of the research (e.g., https://doi.org/10.3390/en14061520 ; https://doi.org/10.3390/en15062245 )
Author Response
Dear reviewers!
We’d like to express our gratitude towards your work. Your comments and corrections helped us to clarify the work and to make it much better.
Authors
Here are some answers to your comments, suggestions and corrections:
Comment: Very good paper. The paper doesn’t contain any critical errors. The applied methodology is robust. The rationale for the research is obvious. The obtained results are credible. Neither improvements nor corrections are necessary. The main advantage of the paper is its concrete and compact character. Very good piece of scientific work.
Answer: Thank you
Comment: Minor corrections: Figure 1 – please add reference dimension.
Answer: The following text was added to Fig.1:
The inner diameter of the reactor is 4.8 m
Comment: More references about methods for calculating neutron fluence, reaction rates, and finally dpa in burnup function can be added to provide more wider context of the research (e.g., https://doi.org/10.3390/en14061520 ; https://doi.org/10.3390/en15062245
Answer: Introduction was improved, some references about methods for calculating neutron fluence were added.
Reviewer 2 Report
Comments and Suggestions for Authors
1) The Article title should be modified – it is not clear and easily understandable at the moment.
2) In the Abstract is completely missing any exact results obtained in this research – some of the most important and the most interesting exact results obtained in this research should be added in the Abstract, because at the moment it is not clear what exact novelties and contributions this paper brings into this research field. Paper novelty and its contribution to the existing research field should be clearly highlighted in the Abstract.
3) The Introduction should be notably modified; more recent literature should be added, proper research field current state should be clearly presented, research gaps resolved in this paper should be clearly explained and highlighted at the end of the Introduction. All the mentioned is currently missing in the Introduction – the Introduction is too rough and too simple at the moment.
4) Subsection 2.2. Mechanical tests – the Authors have properly explained used measurement equipment and measurement procedure. However, for some used measured devices are missing the most important parameters (at least accuracy and precision). Also, the Authors should discuss and present the influence of measurement equipment accuracy/precision on the obtained results.
5) The Article should be written in a neutral form – throughout the paper please remove terms “we”, “our” and similar – and use neutral terminology.
6) Conclusion requires several modifications and improvements:
- First of all, the Conclusion cannot contain bullet points only. The Conclusion beginning and end parts are missing.
- In the Conclusion should be added more exact results (along with their explanation) obtained in this research.
- In the Conclusion, there are missing guidelines in further research.
7) Line 339 – what does it mean? A correction or explanation is required.
8) The Authors should properly discuss and explain how the results obtained in this research can be used and which improvements in the real exploitation conditions can be performed using the results obtained in this analysis.
9) List of the References – this List should be notably enlarged and much more recent literature from this research field (not older than 5 years) should be added, especially considering that most of the currently used references are not recent ones. Notable modifications, improvements, and additions related to the References are required.
10) English is readable and understandable, but it should be notably improved throughout the paper. In the paper exist some unclear sentences, some sentences require better word-order, some sentences should be improved from the grammar viewpoint, etc. Notable improvements and modifications related to the English are required.
Final remarks: This is an interesting paper with properly presented and valuable results. However, the paper requires notable improvements and modifications (according to the comments above) to be acceptable for publication.
Comments on the Quality of English Language
English is readable and understandable, but it should be notably improved throughout the paper. In the paper exist some unclear sentences, some sentences require better word-order, some sentences should be improved from the grammar viewpoint, etc. Notable improvements and modifications related to the English are required.
Author Response
Dear reviewers!
We’d like to express our gratitude towards your work. Your comments and corrections helped us to clarify the work and to make it much better.
Authors
Comment: The Article title should be modified – it is not clear and easily understandable at the moment.
Answer: The title was amended. Changes in the structure and mechanical properties of the SAV-1 alloy and structural Fe-Cr-Ni steels after long-term service as core materials in nuclear reactors
Comment: In the Abstract is completely missing any exact results obtained in this research – some of the most important and the most interesting exact results obtained in this research should be added in the Abstract, because at the moment it is not clear what exact novelties and contributions this paper brings into this research field. Paper novelty and its contribution to the existing research field should be clearly highlighted in the Abstract.
Answer: The Abstract was rewritten.
The article presents the results of studies of the degradation of the structure and mechanical properties of the core materials BN-350 fast neutron and research WWR-K re-actors required to justify the service life extending of early-generation power and re-search reactors. Extending the service life of nuclear reactors is a nowadays problem, since most operating reactors are early-generation reactors that have exhausted their design lifespan. The possibility of extending the service life is largely determined by the condition of the structural materials of the nuclear facility, i.e. their residual resource must ensure safe operation of the reactor. For the SAV-1 alloy, structural material of the WWR-K reactor, studies were conducted on witness samples which were in the active zone during its active operation for 56 years. It was found that yield strength and tensile strength of the irradiated SAV-1 alloy decrease on 24-48% and relative elongation decreases by ~2% compared to unirradiated alloy. Inside the grains and along their boundaries, there are particles of secondary phases enriched with silicon, which is typical for aged aluminum alloys. For irradiated structural steels of power reactors, studied at 350-450C, hardening and a damping nature of creep were revealed, caused by dispersion hardening and the Hall-Petch effect.
Comment: The Introduction should be notably modified; more recent literature should be added, proper research field current state should be clearly presented, research gaps resolved in this paper should be clearly explained and highlighted at the end of the Introduction. All the mentioned is currently missing in the Introduction – the Introduction is too rough and too simple at the moment.
Answer: The Introduction was modified. Resent literature were added.
Comment: Subsection 2.2. Mechanical tests – the Authors have properly explained used measurement equipment and measurement procedure. However, for some used measured devices are missing the most important parameters (at least accuracy and precision). Also, the Authors should discuss and present the influence of measurement equipment accuracy/precision on the obtained results.
Answer: The following were added to the text: The limits of the permissible relative error of the force and deformation sensors were 0.5%, the discreteness of the digital reading device was 0.005% of the nominal load of the force sensor. <...>, the error in determining the temperature of the working part of the sample did not exceed ± 2°C.
Comment: The Article should be written in a neutral form – throughout the paper please remove terms “we”, “our” and similar – and use neutral terminology.
Answer: The text of the article was amended where it was necessary and double-checked.
Comment: Conclusion requires several modifications and improvements: - First of all, the Conclusion cannot contain bullet points only. The Conclusion beginning and end parts are missing. - In the Conclusion should be added more exact results (along with their explanation) obtained in this research. - In the Conclusion, there are missing guidelines in further research.
Answer: The conclusion was rewritten according to your suggestions.
Comment: Line 339 – what does it mean? A correction or explanation is required.
Answer: This line was deleted
Comment: The Authors should properly discuss and explain how the results obtained in this research can be used and which improvements in the real exploitation conditions can be performed using the results obtained in this analysis.
Answer: The discussion and explanation were added to the discussion section and conclusion.
Comment: List of the References – this List should be notably enlarged and much more recent literature from this research field (not older than 5 years) should be added, especially considering that most of the currently used references are not recent ones. Notable modifications, improvements, and additions related to the References are required
Answer: Some resent articles were added to the text and subsequently to the List of the References
Comment: English is readable and understandable, but it should be notably improved throughout the paper. In the paper exist some unclear sentences, some sentences require better word-order, some sentences should be improved from the grammar viewpoint, etc. Notable improvements and modifications related to the English are required.
Answer: The text of the article has been double-checked. We’ve done our best to improve the English.
Reviewer 3 Report
Comments and Suggestions for Authors
The paper presents experimental results on core materials of the a reactor after long term operation. The paper is well structured, and the research significance is good. Therefore, Reviewer suggests of considering the paper for the publication. However, a revision should be made according to the comments provided.
Point 1. To date in literature, research groups are addressing efforts in proposing methodologies for estimating the residual service life of a construction, considering also uncertainties in this evaluation. Among the others Authors may consider the following works:
- org/10.1016/j.istruc.2024.107837
- org/10.1016/j.strusafe.2014.10.003
Point 2. Figure 3. It should be better to report also the non- irradiated alloy stress-strain relationship in order to better compare the results.
Point 3. Figure 6. Please, consider also the values of the non-irradiated alloy for improving the comparison.
Point 4. Conclusion should be rewritten in a discursive way reporting the main outcomes of the work submitted.
Author Response
Dear reviewers!
We’d like to express our gratitude towards your work. Your comments and corrections helped us to clarify the work and to make it much better.
Authors
Comment: The paper presents experimental results on core materials of the reactor after long term operation. The paper is well structured, and the research significance is good. Therefore, Reviewer suggests of considering the paper for the publication. However, a revision should be made according to the comments provided.
Answer: Thank you
Comment: Point 1. To date in literature, research groups are addressing efforts in proposing methodologies for estimating the residual service life of a construction, considering also uncertainties in this evaluation. Among the others Authors may consider the following works: org/10.1016/j.istruc.2024.107837 org/10.1016/j.strusafe.2014.10.003
Answer: The links were added to the text: There are several methods for assessing the possibility of extending the service life of various structures subject to constant or changing stresses, external and internal influences, environmental factors, etc. [4,5]
Comment: Point 2. Figure 3. It should be better to report also the non- irradiated alloy stress-strain relationship in order to better compare the results.
Answer: Stress-strain relationship for non- irradiated alloy was added to Figure 3. Some minor amendments were added to the text.
Comment: Point 3. Figure 6. Please, consider also the values of the non-irradiated alloy for improving the comparison.
Answer: Unfortunately, we didn’t have data on non-irradiated samples. Although same comparisons can be made with data from this article https://doi.org/10.1007/978-1-4020-5903-2_22
This text was added: These results can be compared with properties of unirradiated steels presented in [26].
Comment: Point 4. Conclusion should be rewritten in a discursive way reporting the main outcomes of the work submitted.
Answer: The conclusion was rewritten according to your suggestions.
Reviewer 4 Report
Comments and Suggestions for Authors
This manuscript experimentally investigated the mechanical behaviors of Sav-1 alloy after long-term irradiation. The aging of the materials was focused. The microstructures were observed after mechanical tests and creep tests at various high temperatures. The residual properties of the alloy were evaluated and the new lifespan was predicted. Generally, this research is interesting and can provide some new insights on the extension of lifespan in nuclear reactors. Some improvements are necessary for clarity and logical presentation. My particular comments are as follows:
- Title: Basically, the current title does not have clear meanings and is not smooth.
- Microstructures were observed at different working conditions. But what is the linkage between microstructure change with the change of mechanical properties? How is the residual resource of the steel calculated?
- Can a relationship be established on the residual resource and the irradiation dose, time et al.?
- English and presentations should be significantly improved for clarity and highlighting novelty. The grammatical errors should be avoided.
Comments on the Quality of English Language
Need significant improvements.
Author Response
Dear reviewers!
We’d like to express our gratitude towards your work. Your comments and corrections helped us to clarify the work and to make it much better.
Authors
Comment: This manuscript experimentally investigated the mechanical behaviors of Sav-1 alloy after long-term irradiation. The aging of the materials was focused. The microstructures were observed after mechanical tests and creep tests at various high temperatures. The residual properties of the alloy were evaluated and the new lifespan was predicted. Generally, this research is interesting and can provide some new insights on the extension of lifespan in nuclear reactors. Some improvements are necessary for clarity and logical presentation. My particular comments are as follows:
Answer: Thank you
Comment: Title: Basically, the current title does not have clear meanings and is not smooth.
Answer: The title was amended. Changes in the structure and mechanical properties of the SAV-1 alloy and structural Fe-Cr-Ni steels after long-term service as core materials in nuclear reactors
Comment: Microstructures were observed at different working conditions. But what is the linkage between microstructure change with the change of mechanical properties? How is the residual resource of the steel calculated?
Answer: The issue of the relationship between microstructure and the properties of structural materials are discussed in details in https://doi.org/10.1007/978-1-4939-3438-6. We added this work in introduction, along with some comments.
Comment: Can a relationship be established on the residual resource and the irradiation dose, time et al.?
Answer: The work considered only one of the factors reducing the resource of materials, namely the influence of irradiation (neutron fluence in reactor relevant environment) on strength characteristics. In addition, the residual resource is affected by corrosion of the material, which can be significant.
Links http://dx.doi.org/10.1007/978-1-4020-5903-2_18 and DOI:10.1134/S2075113321030400 were added.
The following sentences were added to the text: An important factor limiting the resource, in addition to the degradation of the mechanical properties of structural materials, is corrosion damage [29, 30].
Comment: English and presentations should be significantly improved for clarity and highlighting novelty. The grammatical errors should be avoided.
Answer: The text of the article has been double-checked. We’ve done our best to improve the English.
Round 2
Reviewer 2 Report
Comments and Suggestions for Authors
The Authors have performed all mentioned corrections/modifications/improvements during the revision process in the best possible (or at least satisfactory) way.
Now, after revision, there are no more concerns related to this paper.
The paper should be accepted and published in the presented (revised) form.
Reviewer 4 Report
Comments and Suggestions for Authors
The authors have addressed my comments. I have no more technical comment.